# AI modeling for outbreak prediction: A graph-neural-network approach for identifying vancomycin-resistant enterococcus carriers

Gregor Donabauer[1,2], Anca Rath[1], Aila Caplunik-Pratsch[1], Anja Eichner[1], Jürgen Fritsch[1], Martin Kieninger[3], Susanne Gaube[4], Wulf Schneider-Brachert[1], Udo Kruschwitz[2], Bärbel Kieninger [1]*

**1** Department of Infection Prevention and Infectious Diseases, University Medical Center Regensburg, Regensburg, Germany, **2** Information Science, University of Regensburg, Regensburg, Germany, **3** Department of Anesthesiology, University Medical Center Regensburg, Regensburg, Germany, **4** UCL Global Business School for Health, University College London, London, United Kingdom

* baerbel.kieninger@ukr.de

## Abstract

The isolation of affected patients and intensified infection control measures are used to prevent nosocomial transmission of vancomycin-resistant enterococci (VRE), but early detection of VRE carriers is needed. However, there are still no standard screening criteria for VRE, which poses a significant threat to patient safety. Our study aimed to develop and evaluate an artificial intelligence (AI)-based approach for identifying and predicting of at-risk patients who could assist infection prevention and control staff through a human-in-the-loop approach. We used data from 8,372 patients, combining more than 125,000 movements within our hospital with patient-related information to create time-dependent graph sequences and applied graph neural networks (GNNs) to classify patients as VRE carriers or noncarriers. Our model achieves a macro F1 score of 0.880 on the task (sensitivity of 0.808, specificity of 0.942). The parameters with the strongest impact on the prediction are the codes for clinical diagnosis (ICD) and operations/procedures (OPS), which are integrated as high-dimensional patient node features in our model. We demonstrate that modeling a "living" hospital with a GNN is a promising approach for the early detection of potential VRE carriers. This proves that AI-based tools combining heterogeneous information types can predict VRE carriage with high sensitivity and could therefore serve as a promising basis for future automated infection prevention control systems. Such systems could help enhance patient safety and proactively reduce nosocomial transmission events through targeted, cost-efficient interventions. Moreover, they could enable a more effective approach to managing antimicrobial resistance.

## Author summary

The rise of antibiotic-resistant bacteria, particularly vancomycin-resistant enterococci (VRE), poses a significant challenge in hospitals worldwide. VRE, which naturally reside

**Data availability statement:** The test data set is a historical data set of our hospital and may not be published for data protection reasons. The implementations of our approach are publicly available via GitLab and can be accessed via the following link: https://git.uni-regensburg.de/dog21258/vre-outbreak-detection.git

**Funding:** The author(s) received no specific funding for this work.

**Competing interests:** The authors have declared that no competing interests exist.

in the human gut, can spread easily in healthcare settings, primarily through contact and contaminated surfaces. These bacteria are highly resilient, surviving for months in dry environments, making them difficult to control and eradicate. Certain patients, such as those with central venous catheters or Clostridioides difficile infections, are more susceptible to VRE colonization, which can lead to severe bloodstream infections. To combat this, hospitals need effective infection prevention and control strategies. However, detecting and isolating VRE carriers is challenging, especially when many carriers are asymptomatic and undetected. Current screening methods are limited, and there is no consensus on the best approach. In response, our research introduces a novel deep learning model that uses a heterogeneous graph neural network to predict undetected VRE carriers in hospitals. This model integrates various patient data and transmission pathways, offering a dynamic and comprehensive representation of the hospital environment. By embedding different spatial entities into the graph and using advanced temporal learning techniques, our approach aims to improve early detection of VRE carriers, thereby reducing the risk of further infections.

## Introduction

Given the global rise in antibiotic resistance, hospitals must develop effective strategies to combat nosocomial infections caused by multidrug-resistant organisms. According to the WHO Bacterial Priority Pathogen List [1], vancomycin-resistant enterococci (VRE) are a significant threat to patient safety and causing a growing burden of disease. In Europe. A rising number of reported enterococcal bloodstream infections (BSI; +57.3%) – and especially of VRE BSI (+80.0%) – was reported between 2017-2021 [2]. Moreover, in 2019, 200,000 deaths worldwide were attributable to or associated with VRE [3].

VRE are Gram-positive cocci found in the human gut. Infections - like bloodstream or urinary tract infections - commonly occur subsequent to colonization and lead to prolonged hospitalization, increase patient suffering and significantly worsen the prognosis of affected patients [4]. A European systematic review analyzing data over a 10-year period reported a mortality rate of 33.5% [5,6]. Certain patient characteristics such as bone marrow transplant, central venous catheter, bedsores, and concurrent *Clostridioides difficile* infection, increase the susceptibility of patients to VRE [7]. Therefore, the development of effective strategies to slow down the spread of VRE and prevent their nosocomial transmission is crucial to avoiding additional risks, particularly for these critically ill patients.

Our current knowledge is that nosocomial VRE transmission occurs mainly through direct contact or through fomites, with healthcare workers' hands being the primary vector. VRE are highly stable in the environment and can survive for months in dry conditions [8–14]. When surfaces are not properly decontaminated, they act as permanent reservoirs for VRE, affecting not only the initial patient but also subsequent patients in the same room or nearby [15–17]. This knowledge has led to the consideration that reduction of shared surfaces between VRE colonized and non-colonized patients – hence contact isolation – may reduce the number of nosocomial transmissions.

A major challenge to this approach – and generally in rapid infection prevention and control (IPC) - is the high number of unknown VRE carriers who can release bacteria into their environment [4,18] but are not subjected to special hygiene and isolation measures, facilitating further transmission. Currently, there is no consensus or sufficient evidence on the use of the VRE screening criterion as an evidence-based measure to prevent VRE spread in hospitals [19,20]. Additionally, cost-benefit analyses have shown that targeted screening is

a more practical approach for VRE detection [21]. Therefore, given the complexity of transmission pathways and VRE patient identification, innovative approaches are needed for early predictive detection of potential VRE carriers to prevent subsequent colonization or infection of other patients.

Recently, artificial intelligence (AI) methods have received attention in various medical tasks due to their ability to automatically extract insights from complex data in the field [22–24].

Examples include training convolutional neural networks (CNNs) to predict cardiac amyloidosis-suggestive uptake from scintigraphy images [25], detecting breast artery calcification from mammograms [26], and estimating carotid artery diameter from ultrasound data [27]. These examples demonstrate how AI can be leveraged for data analysis and pattern recognition in medical applications.

However, beyond analytical capabilities, the interaction between humans and AI is particularly important in medical contexts. Factors such as data privacy, security, and trust perception play a crucial role in the successful adoption of AI-driven solutions. Ensuring that AI tools are interpretable, transparent, and aligned with clinical decision-making processes is essential for their integration into real-world healthcare settings [28,29].

Additionally, techniques from natural language processing (NLP) have recently attracted interest in the field. Examples include early diagnosis of periodontitis using the language model BERT [30], or evaluating the performance of large language models (LLMs) on triage and diagnostic accuracy given clinical case vignettes [31] and clinical information extracting from health records [32].

In the domain of epidemiology, where our work is located, graphs and graph neural networks (GNNs) have drawn particular interest for their capabilities in modeling epidemics and virus outbreaks [33,34]. Graphs, or networks, offer a unique way to model interactions between components in a system [35]. Generally, a graph can simply be interpreted as a collection of objects (i.e., nodes) alongside a set of interactions (i.e., edges) between pairs of these objects [36]. For example, studies have used graph data structures to investigate spatio-temporal settings for predicting the course of epidemic diseases such as influenza or COVID-19 [37–40]. These types of techniques have also been used for the automatic detection of outbreaks of VRE and multidrug-resistant Enterobacterales (MRE). For example, Van Niekerk *et al*. worked with a proprietary dataset to develop a spatiotemporal graph model with nodes as wards and patients as edges to detect VRE colonization at the ward level using a rule-based approach [41]. Furthermore, Gouareb *et al*. modeled temporally evolving information in homogeneous graphs to retrospectively classify patients at risk of MRE infection [42].

However, both studies failed to consider the detailed heterogeneous hospital environment and patient movement data, limiting nodes inside the graphs to a single type, such as either wards or patients. This limitation is critical, as VRE spread is influenced by surface contamination and patient interactions. In addition, Van Niekerk *et al*. rely on statistical models that are limited in their ability to learn nonlinear relationships in the data, while Gouareb *et al*. do not consider the temporal course of interactions and focus on MREs in general ignoring their different biological properties.

Going beyond these limitations, we introduce a deep learning model utilizing a heterogeneous GNN to predict undetected VRE carriers in hospitals. The model combines multidimensional patient data with possible transmission pathways, incorporating both direct and indirect interactions over time (Fig 1). Building on prior research [41,42], our approach involves two key innovations: (1) representing and embedding various spatial entities—such as organizational units, wards, rooms, and beds—into a heterogeneous graph structure, and (2) employing advanced temporal graph learning techniques for a comprehensive temporal

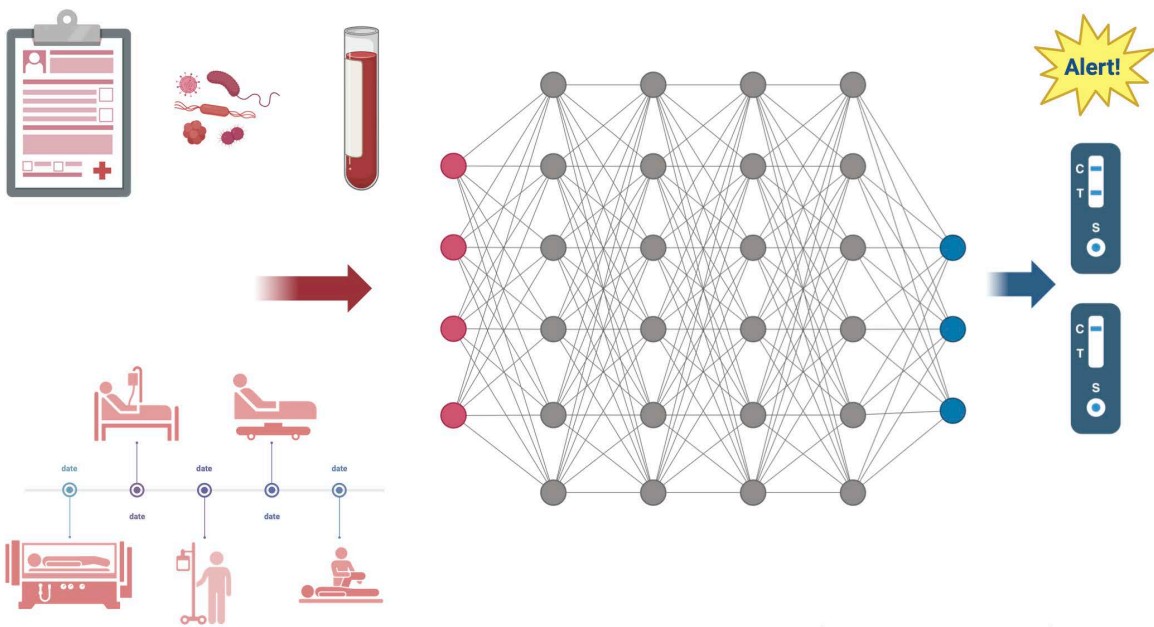

(Created with BioRender.com)

**Fig 1. The idea behind the learning model presented in this paper is to link as much information as possible that is available as a standard in hospital systems in a time-dependent graph neural network. The output is a classification: Would the patient be positive or negative in a possible test for vancomycin resistant enterococci?.**

representation. In the following, we refer to the first stage of development as the "static model" and the second stage as the "dynamic model".

## Results

### Basic analysis

To train our deep learning models and evaluate the approach, we utilized a historical dataset from one year (2019, deliberately chosen before the COVID-19 pandemic) from our tertiary care hospital. This dataset includes data from 8,372 patients (sex: 5,128 males, 3,244 females; age distribution: 0-10 years: 55, 11-20 years: 59, 21-30 years: 327, 31-40 years: 477, 41-50 years: 570, 51-60 years: 1,328, 61-70 years: 1,969, 71-80 years: 1,913, 81-90 years: 1,462, 91-100 years: 208, 101-110 years: 4), representing approximately one-third of the patients hospitalized in our facility during 2019. The dataset contains 125,947 interactions of these patients with the hospital environment, such as admissions, discharges, transfers between departments and rooms for examinations or treatments. The patients interacted with 39 departments, 606 rooms, 770 beds, and 150 organizational units. Fig 2 displays the heterogeneous graph of the hospital, which forms the basis of our models, once without patients (A) and once with the patient movement data for a single day (B).

For training, validation, and testing of our classification approach all patients in the dataset were labeled VRE-positive or VRE-negative. Given that this labeling is based on microbiological test results which are unavailable for all patients in the dataset, some patients are also treated as unknown. Using our defined labeling schema, the distributions for the static model were 247 positive, 801 negative, and 7,328 unknown instances. The distributions for the dynamic model were 53,760 positive, 81,809 negative, and 708,064 unknown instances. The high numbers in the dynamic setting result from the fact that for this model the hospital

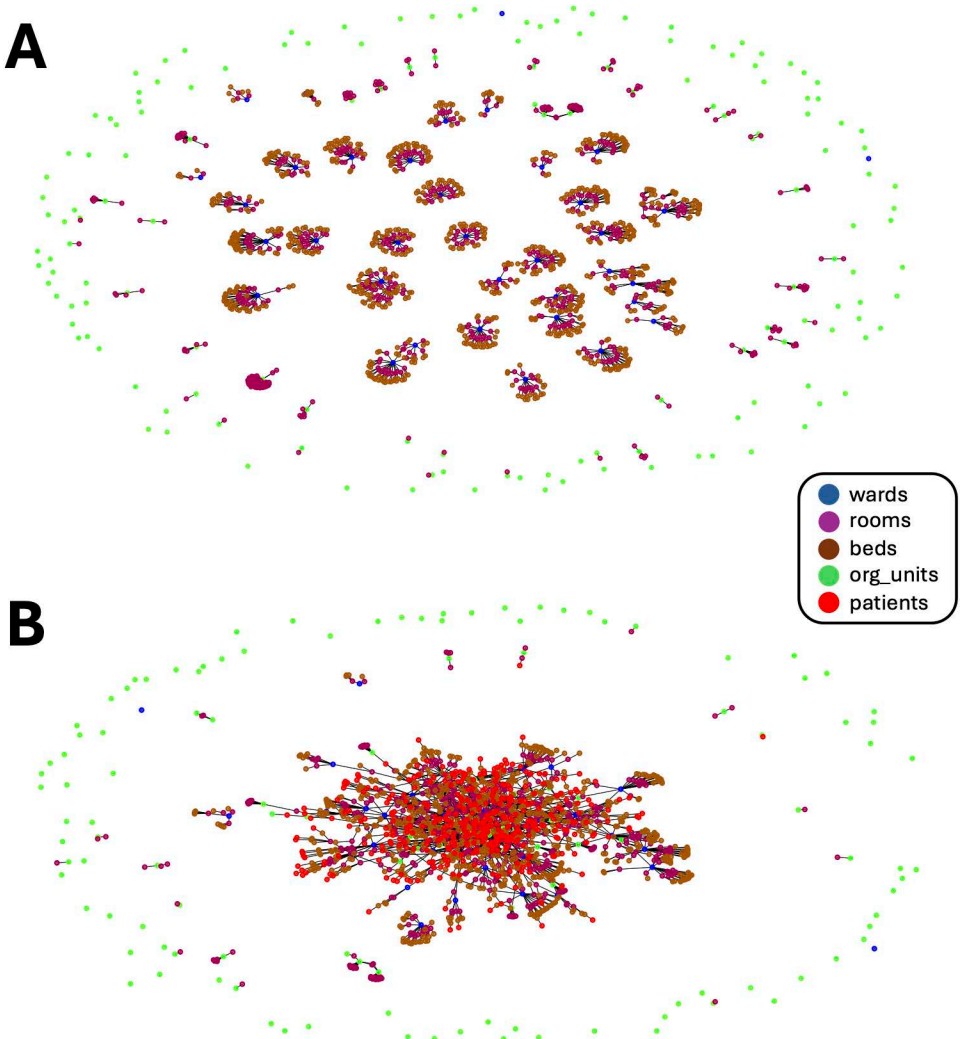

**Fig 2. (A): Graph of the "uninhabited" hospital:** Each of the 39 wards is represented by a node (blue), as are all 606 rooms (magenta), 770 bedplaces (brown) and 150 organizational units (org_unit) (green). The edges between the nodes show the links (room, on, ward), (room, on, org_unit) and (bed, in, room). **(B): Graph of the "living" hospital:** In addition to the nodes shown in (A), this graph contains the patients as a further type of node (red) and the new edges (patient, on, ward), (patient, in, room), (patient, in, bed), and (org_unit, visits, patient), but only for one day (here day 100).

graphs are generated on a daily basis, each containing all patients hospitalized on the respective day. The labels assigned to the patient nodes in the separate graphs are also determined on a daily basis. Fig 3 shows this distribution on a daily basis.

## Model performance

To enhance the expressiveness of the GNN, we assigned node features to the patient nodes and evaluated different feature combinations (no features, one of the feature groups, combination of features, all features) to determine their relevance. Table 1 presents the macro F1 scores, sensitivity, and specificity for both the static model (average results using five-fold cross-validation: a proportion of all patients is classified here) and the dynamic model (average results over five runs distributed equally throughout the year: all patients hospitalized during the test period

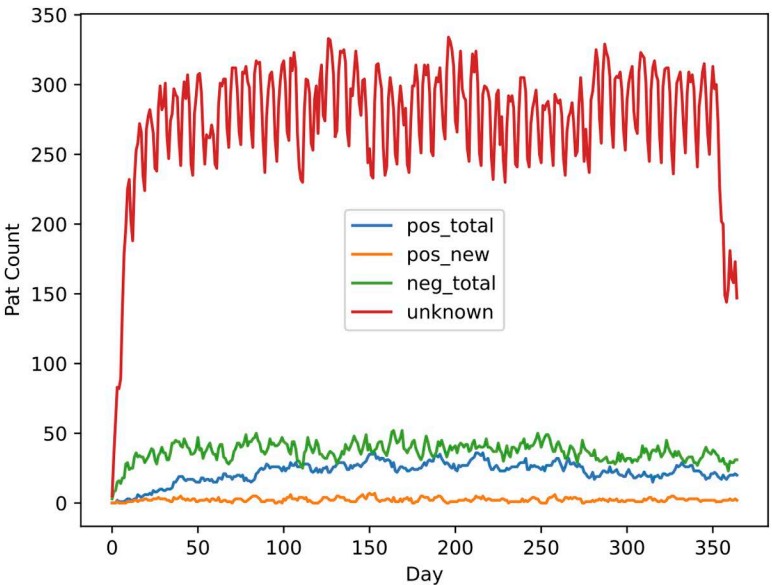

**Fig 3. Patient labels over time:** on each day, the respective values represent the absolute number of patients who fell in each of the four included categories. pos_total = all positive cases known on that day; pos_new = all new positive cases added to the total positive cases on that day; neg_total = all negative cases known on that day; unknown = not falling in either category that day.

**Table 1. Macro F1 scores, sensitivity, and specificity values for the static and dynamic models, based on different configurations of patient node features.**

| Features | Static model | | | Dynamic model | | |
|---|---|---|---|---|---|---|
| | F1 | Sensitivity | Specificity | F1 | Sensitivity | Specificity |
| no features | 0.497 | 0.848 | 0.401 | 0.670 | 0.473 | 0.865 |
| age and sex | 0.452 | **0.864** | 0.327 | 0.683 | 0.557 | 0.805 |
| level of care, information about falls and open wounds | 0.569 | 0.568 | 0.641 | 0.700 | 0.595 | 0.804 |
| Braden (pressure ulcer risk) | 0.487 | 0.760 | 0.419 | 0.706 | 0.539 | 0.863 |
| bedsore | 0.508 | **0.864** | 0.409 | 0.680 | 0.532 | 0.826 |
| bedsore_admission | 0.500 | 0.808 | 0.424 | 0.682 | 0.539 | 0.823 |
| Barthel (disability assessment) | 0.554 | 0.640 | 0.576 | 0.665 | 0.497 | 0.830 |
| Barthel extended | 0.519 | 0.760 | 0.484 | 0.686 | 0.539 | 0.827 |
| days since admission | NA | NA | NA | 0.747 | 0.649 | 0.839 |
| clinical chemistry data from blood | 0.531 | 0.792 | 0.469 | 0.710 | 0.632 | 0.789 |
| ICD codes (diagnoses) | 0.670 | 0.688 | 0.733 | 0.853 | 0.753 | 0.937 |
| OPS codes (operations, procedures) | 0.665 | 0.688 | 0.726 | 0.881 | **0.820** | 0.933 |
| ICD Codes + OPS Codes | **0.693** | 0.632 | **0.796** | **0.885** | 0.796 | **0.958** |
| all features combined | 0.685 | 0.624 | 0.788 | 0.880 | 0.808 | 0.942 |

are classified here). The scores in each row are reported based on results obtained using the individual features for training and testing listed in the "Features" column. Fig 4 provides a graphical comparison of the F1 scores between the static and dynamic models.

The results clearly demonstrated the superiority of the dynamic model as well as the positive impact of incorporating specific patient features such as OPS codes (codes for operations and procedures) and ICD codes (codes for diagnosis), which significantly improved prediction quality. When no patient information was used at all and only the movement and relation

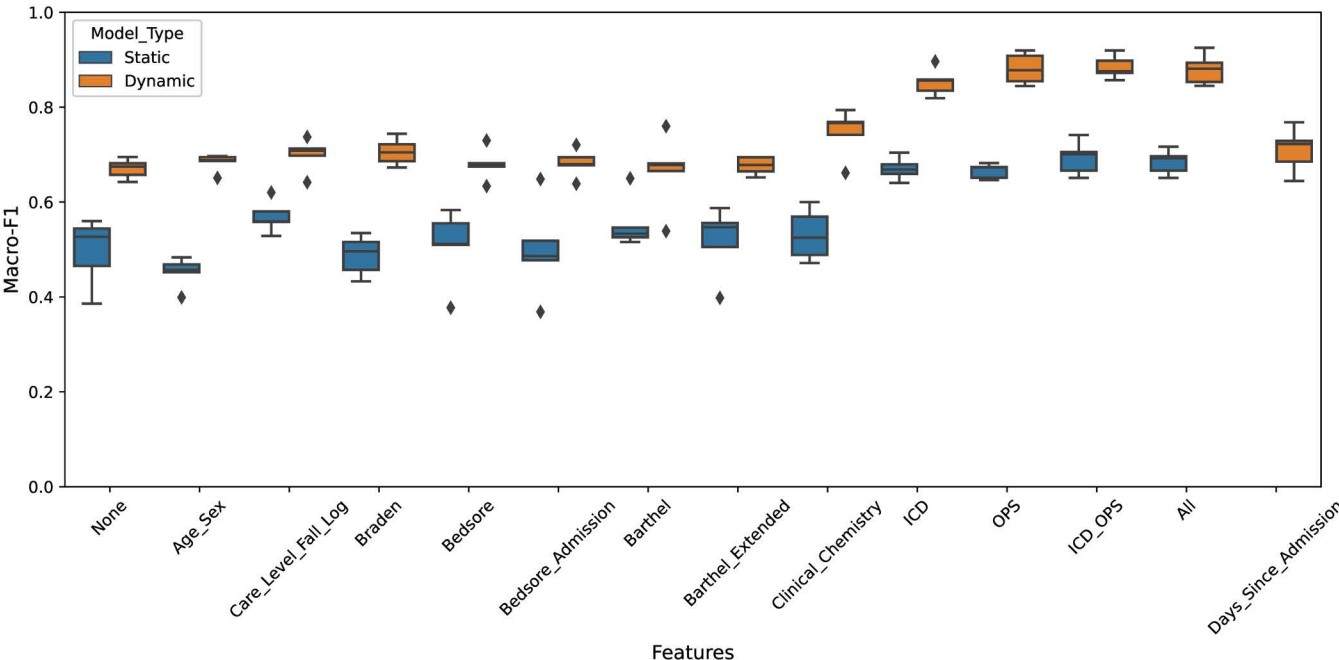

**Fig 4. Performance comparison (macro F1-score) of the static and dynamic model on different features as boxplot.**

network represented by the graph was relied upon, the static model achieved an F1 score of 0.497 (standard deviation (STD) STD=0.064), while the dynamic model predictions resulted in a significantly greater F1 score of 0.670 (STD=0.019). In contrast, when all the available patient features were combined, the static model achieved an F1 score of 0.685 (STD=0.023) while the dynamic model resulted in a significant improvement of roughly 20% with a F1 score of 0.880 (STD=0.029), and the values for sensitivity and specificity also increased markedly. In general, significance tests showed that the dynamic model consistently achieved a significantly better F1 score than did the static model with the same set of features in all setups.

When analyzing the impact of the features, it can be seen that including OPS codes (F1 score=0.881, STD=0.029) and ICD codes (F1 score=0.853, STD=0.026) has the highest positive impact on performance, resulting in a significantly improved F1 score in the dynamic model compared to having no features at all (no features vs. OPS codes: $p_{F1}$=0.008; no features vs. ICD codes: $p_{F1}$=0.008; no features vs. all: $p_{F1}$=0.008). S1 Table includes F1 scores; sensitivity; specificity; precision; recall; accuracy; negative predictive value; positive predictive value; standard deviations from cross-validation or time-curve shifts; p-values for all evaluated model setups and feature combinations. When comparing the dynamic model with all features pairwise against the dynamic model with ICD codes, the model with OPS codes, and the model with both ICD and OPS codes, no statistically significant differences were observed (all features vs. OPS codes: $p_{F1}$=1.000, power = 0.050; all features vs. ICD codes: $p_{F1}$=0.421, power = 0.273; all features vs. OPS and ICD codes: $p_{F1}$=0.841, power = 0.058). However, the statistical power of the analysis is limited by the small sample size (five), suggesting that these results should be interpreted with caution. Nonetheless, the data imply that any potential differences, if present, are likely to be small.

As the length of the training period of the dynamic model was 30 days, these data were selected to calculate the values discussed thus far and are presented in Table 1. To evaluate how this number influences the model performance we varied it to 60 and 90 days in

combination with all available patient characteristics. However, no significant difference was found in any of the evaluation metrics with 60 days of training (F1 score=0.884 (STD=0.033)) or 90 days of training (F1 score=0.882 (STD=0.026)) (30 days vs. 60 days: $p_{F1}$=0.841, $p_{sensitivity}$=1.000, $p_{specificity}$=0.841; 30 days vs. 90 days: $p_{F1}$=0.841, $p_{sensitivity}$=0.690, $p_{specificity}$=0.690), although the significance of the statistical test is severely limited by the low power.

## Discussion

In this study, we successfully developed an AI-based method that uses hospital movement data within heterogeneous graph structures to predict the putative VRE status of patients. The findings from various studies [4,7,10–17] indicating that VRE transmission occurs through contact with VRE carriers and surface contamination inspired us to model hospitals as nodes and edges, thereby replicating transmission routes.

The simplest static model, excluding features such as OPS and ICD codes, showed good sensitivity but low specificity (sensitivity=0.848 (STD=0.099); specificity=0.401 (STD=0.119)). This means that too many cases are falsely classified as positive. By integrating additional features, this model became more precise and accurate (all patient features; sensitivity=0.624 (STD=0.041); specificity=0.788 (STD=0.027)). A fundamental issue with this model is the distortion of the data representation, as it models spatial dependencies that do not exist in reality. For instance, in our model, patients hospitalized at different times within the year are indirectly connected through shared rooms and wards, creating links across these nodes despite months of separation, thereby introducing noise into the data. This means that this model estimates VRE risk per ward/room averaged over the entire observation period.

The improvement from the static to the dynamic model through the integration of temporal information flow is based on enhanced representation of patient histories. The GNN can now learn that certain combinations of diagnoses or procedures occurring in specific temporal patterns lead to a change from VRE negative to VRE positive. Additionally, the time dependence of VRE outbreaks in wards was resolved, accounting for differences in VRE risk per ward/room over time. This is reflected in an average improvement of approximately 20% in all evaluated metrics (see S1 Table) compared to the static model in the all-features setup, which is substantial.

The key predictors of VRE status are diagnosis and procedure codes (OPS: sensitivity=0.753, specificity=0.937; ICD: sensitivity=0.820, specificity=0.933; OPS and ICD: sensitivity=0.796, specificity=0.958), which offer more extensive data than individual features such as bedsore or care level, which were identified as risk factors by Meschiari *et al.* [7], for example. Additionally, incorporating these individual characteristics does not yield significant improvements in model performance when ICD and OPS codes are already included, indicating that combining all available patient features does not necessarily result in the best performance. Given the high dimensional input data, the results also highlight the model's ability to extract relationships from large datasets, demonstrating the efficacy of AI and neural networks. An analysis of the training period indicated that 30 days was sufficient for robust results (sensitivity=0.808, specificity=0.942), with additional days having minimal impact (60 days: sensitivity=0.811, specificity=0.946; 90 days: sensitivity=0.816, specificity=0.938).

Due to the lack of standardized methods for identifying patients at-risk of VRE infection, no established baselines are currently used in hospitals, which limits direct comparison with our findings. Comparing our setup to existing work from Van Niekerk *et al.* [41] and Gouareb *et al.* [42], who classified patients retrospectively after infections occurred, their approaches correspond to our static model. In contrast, our dynamic model approach not only classifies existing VRE carriers but also predicts patients with newly acquired VRE colonizations up to three days before to proven VRE diagnosis. Additionally, our heterogeneous graphs include

all available spatial units — such as organizational units, wards, rooms, and even beds — and thus account for differences in node and edge types compared to homogeneous settings in related work, meeting the need for a detailed model of the environment to track VRE transmission chains [15–17]. In addition, we offer a comprehensive temporal representation by leveraging recent methods in temporal graph learning. This approach is much more comprehensive than the approaches used in these existing studies. When comparing the sensitivity and specificity scores from these two studies to our results, only Gouareb *et al.* [42] reported these metrics, whereas Van Niekerk *et al.* [41] focused on ward-level predictions and reported on the area under the curve (AUC) metric instead. Gouareb *et al.*'s best-performing setup demonstrated comparable performance (sensitivity = 0.806, specificity = 0.966) when addressing a broad range of multidrug-resistant bacteria. However, different bacteria have variable transmission pathways, patient populations at risk and pathogenicity, making the feasibility of this score insufficient for the evaluation of VRE in particular due to a lack of precision. While our experimental setup is different to this work (we focus on VRE as one type of multi-resistant bacteria and predict infections up to three days in advance) it places our findings in line with prior research and considers the specific insight into the characteristics of this particular pathogen.

Future work should focus on deploying our system in a "live" setting within the ICP workflow, which leads us to discuss practical implications and alignment with existing hospital processes. The question of what constitutes an ideal interface between a human and an AI application has attracted a lot of attention recently and the answer to that question clearly depends on each individual use case [43]. As outlined in the Methodology, a GPU with 6GB of VRAM is sufficient to retrain our proposed model daily (even with increasing graph complexity, e.g., a higher number of patients) to maintain an up-to-date predictor. The most challenging part of setting up the system is integrating existing hospital information systems, which house the relevant data, with the graph representation upon which our algorithm is based. This might require manual effort by technical and medical experts in individual environments. However, once the relevant data are matched, the remaining steps of model training and infection prediction are straightforward. After deployment, another challenge is making the system easily accessible to IPC personnel. The ideal solution may involve developing a user-friendly interface that bridges hygiene experts responsible for screening with the algorithm predicting at-risk patients. However, developing such an interface requires additional effort, such as identifying an easily accessible and user-friendly presentation of the information or ensuring that there is no misinterpretation of the algorithm's screening recommendations. These can then serve as guidance for but not replace IPC team.

It is important for us to emphasize that our project, made possible only through the digitization of patient records, aims to give this documentation a broader purpose by actively utilizing the resulting dataset to improve diagnostics, therapy, and patient care. In the future, we hope to contribute to evidence-based data utilization for the benefit of patients and to support efforts in combating multidrug-resistant pathogens and antibiotic resistance.

## Limitations

Overall, we highlight that this study serves as a proof of concept, which entails in a number of limitations that need to be addressed in future research. Below, we outline these limitations in detail and provide an outlook on how we plan to address them in future work.

Although our dataset included 8,372 patients, only 247 were microbiologically identified as definite VRE carriers, resulting in a limited number of positive samples for training the neural network. The division of patients into training, validation, and test sets further reduced the number of patients considered. Additionally, our dataset includes data from only a single

hospital, covering only one-third of the patients hospitalized during the one-year observation period. To mitigate this limitation, we employed cross-validation (for the static model) and temporal splits (for the dynamic model) to establish a more robust evaluation setting, demonstrating that our approach yields promising results. However, we emphasize that future research should expand on our work by including a larger and more complete sample of patients over an extended period (e.g., by examining multiple years instead of just one) and incorporating data from additional hospitals to improve the generalizability of our findings.

While our study focused exclusively on VRE, one of the most critical multidrug-resistant bacteria, our approach can be adapted to target other types of bacteria by modifying the target labels. However, its performance in such settings needs to be evaluated to determine whether it is comparable to the results we observed with VRE. For the method to be effectively applied, a certain minimum number of carriers of the microorganism in question must likely be present and known. Otherwise, the system would lack sufficient patterns for learning, which are essential for making reliable predictions. As a result, the method may not be applicable to every potential nosocomial pathogen.

Another limitation of the dataset used in our study is its retrospective nature, which may introduce bias into the modeling process, for example, by including diagnoses that might not be fully available in real-time. This emphasizes the importance of evaluating model performance in predicting VRE carrier status using prospective, real-time data that are automatically limited to information available up to the current day. This would also enable assessing whether performance in such a setting aligns with the results we observed in our study.

As a final and significant limitation, we would like to highlight the uncertainty of the labeling due to the limited sensitivity of VRE screening tests. The detection rate of rectal swabs for identifying VRE carriers has been reported to be as low as 58%, depending on the bacterial density in stool samples [44]. Consequently, some VRE-positive patients in our dataset may have been misclassified as negative, potentially affecting the quality of the labels used for model training. Additionally, early-stage colonization may not be detected by a single test, necessitating serial sampling for accurate identification [45]. This diagnostic uncertainty in screening strategies could impact the model's performance, and future studies should consider its potential influence on predictive accuracy.

## Conclusion

We demonstrated that modeling a "living" hospital as a graph and processing it with a GNN is a promising approach for the early detection of VRE carriers. We successfully identified patients in a historical dataset who were most likely to test positive for VRE within the next three days. If this method could be applied in real time, it could prove to be a powerful alternative for the current unsystematic VRE screening procedures.

Ultimately, this strategy would provide a temporal advantage for IPC teams by enabling the early identification of potential VRE carriers. Since unrecognized carriers can contribute to silent transmission, their timely detection is essential for preventing larger transmission chains. By systematically identifying at-risk patients, our approach lays the groundwork for effective IPC strategies and supports outbreak prevention—reinforcing the idea that outbreak prediction starts with carrier identification.

## Methods

### Ethics approval

The study was approved by and conducted according to the guidelines of the Ethics Committee of the University of Regensburg (approval number 22-3113-104, 27th October 2022).

## Data collection

The first step was to compile a representative dataset from our tertiary care hospital to model the underlying patterns and information for classifying patients as VRE carriers or noncarriers, ensuring that the selected data could be generalized to other hospitals to enable the method to be applied there as well. The year 2019 was chosen as the observation period. The team's IPC experts selected 11 wards (vascular surgery, internal medicine, nephrology, oncology, ophthalmology and surgery) and four intensive care units (ICUs; internal medicine, general and emergency surgery and cardiothoracic surgery) to include patients with a higher and lower risk of infection, respectively. All patients who spent time in one of the wards during the observation period were included.

All other wards were also mapped as nodes in the graph and included through patient contacts. The patient-related information used as patient characteristics in the network included basic patient data, patient movements, and patient features from the patient data management system as well as clinical chemistry data from the laboratory information system. Patient movements, which form the foundation of the underlying graph data structure, can be directly extracted from the patient data management system. As the system contains information about patient transfers between wards and rooms, we can use this information to represent spatial interactions within the graph. By continuously modeling this discrete temporal data, we create a detailed representation of patient movement paths over time. The code U80.30 (*Enterococcus faecium* with resistance to glycopeptide antibiotics) was excluded from the dataset to avoid biasing the model by explicitly providing VRE infection diagnosis. We do not explicitly include microbiological data from the laboratory information system as patient characteristics. However, such information can be to some degree encoded in the ICD codes that are part of the patient features. We will use microbiological data (VRE test results) to assign labels to patients. In the following sections, we further explain how these different data categories were used in our model.

## Model structure

For both the static and dynamic models, we represented the network of hospital elements (wards, rooms, beds, organizational units) and patients as a graph holding spatial relations as well as movement paths across the hospital in one data structure.

**Representation as a Graph.** To model the spatial relationships within the hospital, we created an undirected, unweighted heterogeneous graph [36], implemented using PyTorch Geometric [46]. This graph is characterized by different node sets, which are connected through multiple sets of edges. The graph includes the following sets of nodes: wards, rooms, beds, organizational units (org_unit), and patients. Additionally, edge sets denote relations such as (room, on, ward), (room, on, org_unit), (bed, in, room), (patient, on, ward), (patient, in, room), (patient, in, bed), and (org_unit, visits, patient). Initially, we constructed a static graph model of the hospital, incorporating all spatial units and their connections while excluding patient-related information. This information is derived by first identifying all unique spatial units from the complete list of patient movements extracted from the patient data management system. Relationships between these units can then be identified by analyzing co-occurrences of spatial (sub)levels in the data—for example, room numbers associated with the same ward numbers. Subsequently, we integrated patient-related information into the network via patient nodes and their connections to the hospital nodes. This network represents a detailed history of patient movements and dependencies within the hospital. Each node in the graph is represented by a feature vector. In the case of hospital-related nodes, we used one-hot encodings to serve as unique identifiers. For ward nodes, an

additional binary feature indicates whether the ward treats ICU patients. The patient nodes are characterized by high-dimensional feature vectors that contain extensive medical and personal data (dimensions in brackets), with maximum (max) and minimum (min) values for the clinical chemistry data referring to all values measured during the hospital stay: age (1), sex (1), care level (1), presence of a fall log (1), presence of open wounds (1), diagnosis (4576), Braden scale - pressure ulcer risk (1), bedsore (1), bedsore at time of admission (1), Barthel index - disability assessment score (1), Barthel index extended (1), operation codes (4034), days since admission (1), and clinical chemistry data from blood (Leucocytes min, max; Hemoglobin min, max; Creatinine max; C-reactive protein max; Aspartate transaminase max; Alanine transaminase max; Free hemoglobin min, max; Hemoglobin A1C max) (17). It should be noted that patient nodes do not receive an 'identifier' as a feature, which is relevant for the separation of training, validation and test data in our dynamic model.

**Static and dynamic graph data.** For the static model, as a baseline and to test whether VRE patients can be identified by their links and characteristics within the graph, we modeled all patient movements from the dataset within a single graph. Temporal shifts between movements are ignored in this approach; however, these shifts result in linking patients at the beginning of the observation period to patients at the end of the observation period inside the graph. The patient nodes also contained all available features without specifying the date. In this setting, a proportion of patient nodes distributed over the observation period served as a test set (applying five-fold cross-validation) to evaluate how well predicting VRE-positive or VRE-negative patients worked after training the neural network on all remaining patient nodes.

To realize the idea of an alert system, we incorporated temporality into our dynamic model as follows: To account for temporal dynamics, we created a separate graph for each day, capturing the spatial relations and patient movements occurring on that specific day. Similarly, patient node features included only information observable until the respective day.

To inject spatiotemporal information into the network, we linked consecutive day graphs within n-day-long sequences by introducing new temporal edge sets (n=7, which represents the average number of days patients spend in a hospital in Germany), which aligns with previous work on temporal graph machine learning [47,48]. These edges connected the same nodes present across days, such as nodes from the current day G(T) with those from day G(T-1) and from day G(T-1) with G(T-2). The historical information within each sequence representing day G(T) returned to G(T−7). In this way, the test data could be separated from the data used for learning via a time cutoff.

**Labeling.** As our goal was to classify patients based on their VRE status, we had to assign labels to each patient node in the graph for training and evaluating the neural networks. These labels were derived from microbiology test results and fell into three categories: (0) negative, (1) positive, and (2) unknown. It should be noted that no distinction is made here between VRE colonization and VRE infection. For model training and evaluation, we considered only labels (0) and (1), as the status of the patients labeled (2) could not be reliably determined. The learning scenario is therefore semisupervised. This means that during training, the weights of the neural network are adjusted solely based on the loss incurred when predicting the VRE carrier status of patients labeled as positive or negative. While the model also predicts a positive/negative label for patients we consider unknown, this information is discarded, as we cannot reliably compare these predictions to a ground truth. The same applies to evaluation.

In the static model, labeling is straightforward: a patient is labeled positive if they have at least one positive test result throughout the year. Conversely, a patient is labeled negative if they do not have any positive test results but have at least one negative test result. Patients who do not meet either condition are labeled as unknown.

In the dynamic model, we define the assignment of labels as a function of time as follows: On day T, a patient is labeled positive if they had a positive test result before time T or up to time T+3 (this allows us to predict test results up to three days in the future). Conversely, a patient is considered negative if they never had a positive test result up to time T+3 and simultaneously had a negative test result between time T-2 and T+12 (which allows us to consider patients only negative if they have a recent negative test result for up to two weeks in the future). Patients not falling into either category are labeled "unknown" at time T. Consequently, labels for an individual patient can change over time owing to their dependence on temporal dynamics.

**Neural network.** To determine infection patterns based on the heterogeneous network structures described, we employed an end-to-end GNN. Since graph data structures go beyond traditional inputs such as sequences (e.g., text) or grid structures (e.g., images) they require a different type of encoder than well-defined neural networks such as recurrent or convolutional neural networks [36]. As a solution, GNNs that can operate on graph-structured data have been proposed [36]. In detail, our GNN first reduced the dimensionality of node feature input by using fully connected layers. For spatiotemporal signal aggregation within the graph sequence, we then input these representations through two successive heterogeneous graph transformer layers [49], a GNN layer type that has proven to be effective in other areas of heterogeneous graph learning [50]. Finally, a fully connected layer maps the hidden representations of the patient nodes into the label space.

For model training and evaluation, we perform different data splits based on the respective setup (static versus dynamic):

With respect to the static model, we ran a transductive approach with the full set of patient nodes using fivefold cross-validation. The test set was further divided into validation and test nodes. When setting up the neural network, we used a hidden layer dimension of 512, a batch size of 512, and a learning rate of 0.00001. We trained the model for 200 epochs with early stopping and 70 epochs. These parameter settings have shown robust performance in pre-experimental runs.

In the dynamic model, we used the full graph sequences with all involved patient nodes up to a specific point in time for training. The subsequent 7-day sequences were used for validation, and the following 7-day sequences served as the test set. We ran this setup five times with different temporal shifts. To evaluate how the length of the training period influences performance, we varied the number of training days between 30, 60, and 90. To allow comparability, the validation and test periods were always identical between setups, independent of the number of training days used. We again set the hidden layer dimension to 512 and the learning rate to 0.00001. However, in the temporal setting, one batch consisted of a whole graph sequence representing day T and its context up to T-7, with all patients present on day T forming one batch. Training was conducted for 200 epochs with early stopping and a patience of 10 epochs.

We note that using cross-validation and temporal splits with independent training, evaluation, and test sets is a well-established approach for the robust evaluation of machine learning models in medicine [51,52]. Furthermore, our evaluation extends beyond previous work in the field, which relied on a single fixed train-test split [41] or a single train-evaluation-test split [42].

Both of our models were trained using the Adam optimizer and cross-entropy loss for parameter optimization. Performance was evaluated using metrics from classification and infectiology: precision, recall, macro F1 score (unweighted average of per-class F1 scores, each of which be calculated as 2 × True Positives2 × True Positives + False Positives + False Negatives), accuracy, sensitivity, specificity, negative predictive value, and positive predictive value. All experiments were executed on a single NVIDIA RTX 2060 GPU with 6GB VRAM.

To assess the impact of the different characteristics of the patient nodes on the metrics, the models were trained with different input feature sets. To identify differences between the models (static vs dynamic) and different input feature combinations, the Mann-Whitney U test was used to test for significance (2-sided), with a significance level of 0.05 (SPSS Statistics, Version 29.0.1.0, IBM, Armonk, USA).

## Supporting information

**S1 Table. Further information on the scores of the static and dynamic model, based on different configurations of patient node features.** S1 includes macro F1 scores; sensitivity; specificity; precision; recall; accuracy; negative predictive value; positive predictive value; standard deviations from cross-validation or time-curve shifts; p-values for all evaluated model setups and feature combinations.
(PDF)

## Acknowledgments

We would like to thank Dr Wolfgang Börner for his help in developing the data protection concept and Petra Fest, Anna Saibold, Dr Alexander Wimmer and Doris Mühlbauer for exporting the data from the databases. We would also like to thank Rosemarie Rothe for her help in selecting the parameters.

ChatGPT and DeepL were used to improve readability.

## Author contributions

**Conceptualization:** Susanne Gaube, Wulf Schneider-Brachert, Udo Kruschwitz, Bärbel Kieninger.

**Data curation:** Gregor Donabauer, Bärbel Kieninger.

**Formal analysis:** Gregor Donabauer, Anja Eichner, Bärbel Kieninger.

**Investigation:** Gregor Donabauer, Bärbel Kieninger.

**Methodology:** Gregor Donabauer, Udo Kruschwitz, Bärbel Kieninger.

**Project administration:** Bärbel Kieninger.

**Resources:** Wulf Schneider-Brachert.

**Software:** Gregor Donabauer, Bärbel Kieninger.

**Supervision:** Jürgen Fritsch, Wulf Schneider-Brachert, Udo Kruschwitz.

**Validation:** Anca Rath, Aila Caplunik-Pratsch, Martin Kieninger.

**Visualization:** Gregor Donabauer, Jürgen Fritsch, Bärbel Kieninger.

**Writing – original draft:** Gregor Donabauer, Anca Rath, Bärbel Kieninger.

**Writing – review & editing:** Gregor Donabauer, Anca Rath, Aila Caplunik-Pratsch, Anja Eichner, Jürgen Fritsch, Martin Kieninger, Susanne Gaube, Wulf Schneider-Brachert, Udo Kruschwitz, Bärbel Kieninger.

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
