## [Decision Letter · Decision Letter 0]

13 Jan 2025

PDIG-D-24-00384AI Modeling for Outbreak Prediction: A Graph-Neural-Network Approach for Identifying Vancomycin-Resistant Enterococcus CarriersPLOS Digital Health Dear Dr. Kieninger, Thank you for submitting your manuscript to PLOS Digital Health. After careful consideration, we feel that it has merit but does not fully meet PLOS Digital Health's publication criteria as it currently stands. Therefore, we invite you to submit a revised version of the manuscript that addresses the points raised during the review process. Please submit your revised manuscript within 60 days Mar 14 2025 11:59PM. If you will need more time than this to complete your revisions, please reply to this message or contact the journal office at digitalhealth@plos.org. Please include the following items when submitting your revised manuscript:* A rebuttal letter that responds to each point raised by the editor and reviewer(s). You should upload this letter as a separate file labeled 'Response to Reviewers '. This file does not need to include responses to any formatting updates and technical items listed in the 'Journal Requirements' section below.* A marked-up copy of your manuscript that highlights changes made to the original version. You should upload this as a separate file labeled 'Revised Manuscript with Track Changes '.* An unmarked version of your revised paper without tracked changes. You should upload this as a separate file labeled 'Manuscript '. If you would like to make changes to your financial disclosure, competing interests statement, or data availability statement, please make these updates within the submission form at the time of resubmission. Guidelines for resubmitting your figure files are available below the reviewer comments at the end of this letter. We look forward to receiving your revised manuscript. Kind regards, Sulaf Assi, PhDAcademic EditorPLOS Digital Health  Leo Anthony CeliEditor-in-ChiefPLOS Digital Healthorcid.org/0000-0001-6712-6626 **Journal Requirements:**

**Reviewers' Comments:** Reviewer's Responses to Questions

**Comments to the Author**

Reviewer #1: The paper presents a new approach to predicting VRE carriers using artificial intelligence. The reviewer believes there are some important drawbacks and weakness that should be considered to improve the quality of the paper.

1. A clearer statement of the study's significance or implications for clinical practice should be provided in Abstract. It lacks a brief mention of the potential impact on patient safety or infection control strategies.

2. More context on the global impact of VRE infections, including statistics on morbidity and mortality, to emphasize urgency can be added to Introduction to outlines the problem of VRE.

3. The literature survey could be strengthened by reviewing more recent advancements in AI applications in healthcare beyond VRE. This would help situate the study within a broader research landscape.

4. Certain aspects, such as how patient movement data were collected and processed are unclear. More information on data preprocessing steps would enhance reproducibility.

5. The approach to labeling patients as VRE-positive, negative, or unknown is not fully transparent. A clearer explanation of how unknown statuses were treated in model training and evaluation would improve understanding.

6. The methodology validation techniques used to assess model performance beyond macro F1 scores, including cross-validation or external validation methods are poor.

7. There is a lack of visual aids (e.g., performance metrics graphs) that could enhance comprehension of model efficacy. More figures illustrating key findings would be beneficial.

8. Providing context on how sensitivity and specificity compare to existing methods would add depth.

9. A more thorough acknowledgment of limitations would strengthen the paper's credibility. Besides, suggestions for how hospitals might implement AI tools based on findings would provide practical value.

10. Simplifying language where possible could enhance understanding without sacrificing scientific rigor.

Reviewer #2: The manuscript presents a compelling exploration of how AI, and in specific GNNs, can be employed to address a critical challenge in healthcare, mainly contributing to the early identification of vancomycin-resistant enterococci (VRE) carriers. This research is situated at the intersection of computational modeling and infection prevention, an area of growing importance given the global rise in antimicrobial resistance. By leveraging hospital data to model patient interactions and spatial dynamics, the study offers a data-driven approach to predicting VRE status, with the potential to transform current infection control practices.

Methodological Evaluation: This study introduces a sophisticated application of GNNs to predict the presence of VRE carriers in a hospital setting. Methodologically, the research distinguishes itself by employing two models: a static model that aggregates patient movement data without temporal dynamics, and a dynamic model that integrates temporal graph learning. The dynamic model, which contextualizes patient interactions over time, is particularly noteworthy and outperforms the static model in all evaluated metrics as it showed in the submission. The reliance of this study on time-dependent graph sequences to model a "living hospital" environment is certainly an innovative approach per se, and it addresses a critical gap in infection prevention and control research. The inclusion of heterogeneous data points, such as ward and room locations, patient movements, and ICD/OPS codes, reflects an advanced integration of multidimensional data for predictive modeling. The application of advanced temporal learning techniques further enhances the methodology, offering predictive capabilities that could significantly benefit infection control protocols in real-world clinical settings. However, there are limitations in the methodology that warrant attention, and mainly raising concerns on the generalizability of its outcomes, as the training data were collected and used from a single hospital, and the secondly, the relatively small sample of confirmed VRE cases (247 patients) may constrain the robustness of the findings. While the use of cross-validation partially mitigates these limitations, future iterations of the study should include multi-center datasets and explore external validation to strengthen its applicability.

Scientific Contribution and Value: The manuscript contributes meaningfully to the field of healthcare-associated infection prevention by advancing predictive analytics for VRE carrier identification. The incorporation of GNNs into hospital data modeling helped the study in addressing the inherent complexity of VRE transmission pathways, which involve both direct patient interactions and environmental contamination. The predictive framework, which anticipates VRE status up to three days in advance, is particularly impactful, as it provides infection control teams with a temporal advantage to implement preventive measures. On another note, this research builds upon and extends prior studies by leveraging heterogeneous graph structures that capture detailed environmental variables, such as beds, wards & rooms, and temporal patient trajectories. Unlike these earlier studies, which were limited to retrospective classification or homogenous graph models, the current approach offers a dynamic and forward-looking perspective, positioning it as a potential cornerstone for automated infection prevention systems. Despite its evident value, the study's real-world utility would be significantly enhanced by a more detailed discussion on its scalability and integration into existing hospital workflows. Additionally, the lack of a thorough cost-benefit analysis leaves open questions about its feasibility in resource-constrained settings.

Discussion and Conclusions: The authors articulate the strengths of their model with clarity, emphasizing its capacity to predict VRE status dynamically and its superiority over static models. The discussion effectively contextualizes the findings within the broader literature on AI applications in infection control, demonstrating both the novelty and relevance of the proposed methodology. However, the manuscript could benefit from a more critical evaluation of its limitations, particularly regarding the potential biases introduced by the single-center dataset and the reliance on retrospective data. The authors do acknowledge these limitations but stop short of discussing their implications for broader implementation. A more detailed exploration of the practical barriers to real-time deployment, such as computational demands and integration with electronic health record (EHR) systems, would strengthen the manuscript’s impact. On a different note, the conclusion, while succinct, highlights the study's promise as a tool for early detection of VRE carriers, underscoring its potential to revolutionize current screening practices. By enabling the proactive interventions, this approach could reduce nosocomial transmission rates and improve patient safety. However, the authors should expand their discussion of how this methodology could be adapted to other multidrug-resistant organisms (MDROs) to underline its broader applicability.

In conclusion, the study introduces several groundbreaking innovations: a) integrating of temporal dynamics into graph modeling which represents a significant methodological advancement, enabling the prediction of patient VRE status with unprecedented accuracy; b) the inclusion of diverse data types, be it spatial, temporal, and clinical, captures the complexity of hospital environments and offers a granular view of VRE transmission pathways; c) the ability to forecast VRE status up to three days in advance is a transformative step forward, offering actionable insights for infection control teams. These features collectively position the study at the forefront of AI applications in healthcare-associated infection control.

Recommendation: Based on the strengths and innovative approaches applied and integrated by this study, and despite the highlighted limitations of the manuscript, I recommend accepting this manuscript for publication, after highlighting the limitations of this study.

Reviewer #3: Great paper, in well-described depth as well. It would be helpful to hear more about the "microbiological data" comment which prompted a question in my mind about whether culture data (blood, urine, etc) was included. That would obviously be critical to prevent data leakage in addition to the ICD code excluded, but there is good description elsewhere in the methods section of the other variables included to my satisfaction.

Reviewer #4: Thanks for the opportunity to read this interesting paper.

The authors explored the benefits of artificial intelligence tools to identify and predict patients at risk of VRE carriage. The purpose is of real interest for IPC practitioners and epidemiologists and deals with the tools and data available due to IT.

The paper is easy to read and well-written.

I think the term “outbreak detection” seems a too much because the aim of the study is to detect VRE carriers, not outbreak.

I should be valuable to evaluate the performance of the models on a different year of the hospital or in another hospital. This would reinforce the validity of the models. We're at the feasibility study stage at the moment.

The key predictors found in the study are diagnosis and procedures codes. It's consistent with literature, but it's not the more interesting data to make a real-time predictive tool. These data are often available a few days or weeks after patient discharge.

In table 1, I am not sure to understand some terms and theirs meaning: Braden? Barthel? Bedscore? Clinical chemistry from blood? ... I think this should explain in methods.

In the discussion, I think the contribution of the novel model in real life for IPC should be discussed. As well as the work to do before implementing the model.

---

## [Editor Report · Decision Letter 1]

8 Mar 2025

AI Modeling for Outbreak Prediction: A Graph-Neural-Network Approach for Identifying Vancomycin-Resistant Enterococcus Carriers

PDIG-D-24-00384R1

Dear Dr. Kieninger,

We are pleased to inform you that your manuscript 'AI Modeling for Outbreak Prediction: A Graph-Neural-Network Approach for Identifying Vancomycin-Resistant Enterococcus Carriers' has been provisionally accepted for publication in PLOS Digital Health.

Best regards,

Dhiya Al-Jumeily OBE, PhD

Section Editor

PLOS Digital Health

**Additional Editor Comments (if provided):**

Many thanks for addressing the comments thoroughly and comprehensively. The manuscript is in publishable state.